# Effectiveness of Silver Nanoparticles Deposited in Facemask Material for Neutralising Viruses

**DOI:** 10.3390/nano12152662

**Published:** 2022-08-03

**Authors:** Raúl López-Martín, Imanol Rodrigo, Carlos Ballesta, Armando Arias, Antonio Mas, Benito Santos Burgos, Peter S. Normile, Jose A. De Toro, Chris Binns

**Affiliations:** 1Instituto Regional de Investigación Científica Aplicada (IRICA), 13005 Ciudad Real, Spain; raul.lopez@uclm.es (R.L.-M.); benito.santos@uclm.es (B.S.B.); peter.normile@uclm.es (P.S.N.); joseangel.toro@uclm.es (J.A.D.T.); 2Departamento de Física Aplicada, Universidad de Castilla-La Mancha, 13071 Ciudad Real, Spain; 3Unidad de Biomedicina, CSIC, Universidad de Castilla-La Mancha, 02008 Albacete, Spain; imanol.rodrigo@uclm.es (I.R.); carlos.ballesta@alu.uclm.es (C.B.); armando.arias@uclm.es (A.A.); antonio.mas@uclm.es (A.M.); 4Unidad de Medicina Molecular, Centro Regional de Investigaciones Biomédicas (CRIB), Universidad de Castilla-La Mancha, 02008 Albacete, Spain; 5Escuela Técnica Superior de Ingenieros Agrónomos, Universidad de Castilla-La Mancha, 02006 Albacete, Spain; 6Facultad de Farmacia, Universidad de Castilla-La Mancha, 02071 Albacete, Spain

**Keywords:** nanoparticle, antiviral, aerosol

## Abstract

Cloth used for facemask material has been coated with silver nanoparticles using an aerosol method that passes pure uncoated nanoparticles through the cloth and deposits them throughout the volume. The particles have been characterized by electron microscopy and have a typical diameter of 4 nm with the atomic structure of pure metallic silver presented as an assortment of single crystals and polycrystals. The particles adhere well to the cloth fibers, and the coating consists of individual nanoparticles at low deposition times, evolving to fully agglomerated assemblies in heavy coatings. The cloth was exposed to Usutu virus and murine norovirus particles in suspension and allowed to dry, following which, the infectious virus particles were rescued by soaking the cloth in culture media. It was found that up to 98% of the virus particles were neutralized by this contact with the silver nanoparticles for optimum deposition conditions. The best performance was obtained with agglomerated films and with polycrystalline nanoparticles. The work indicates that silver nanoparticles embedded in masks can neutralize the majority of virus particles that enter the mask and thus increase the opacity of masks to infectious viruses by up to a factor of 50. In addition, the majority of the virus particles released from the mask after use are non-infectious.

## 1. Introduction

The effectiveness of Ag as an antibacterial agent dates back to before antibiotics [1], and in the past twenty years, there has been renewed interest in Ag in the form of nanoparticles for disinfecting materials against antibiotic-resistant bacteria [2,3,4]. Nanoparticles have attributes that increase their effectiveness, including the high surface-to-mass ratio and increased catalytic activity due to the high proportion of active facets [5]. While some studies have found that nanoparticles per se do not give enhanced antibacterial activity beyond that of the available quantity of Ag ions [6], others demonstrate a significant size effect, with the best performance coming from particles with diameters in the range 1–10 nm [7].

There is less work published on the antiviral activity of nanoparticles. The first study was reported in 2005 [8], namely an investigation of the interaction of Ag with HIV virus particles. It showed that the Ag nanoparticles attach to the glycoproteins that cover the virions and are used to dock with and enter cells; thus, entry into cells and infection is inhibited. Since then, the interaction of a range of nanoparticles, including Ag, Au, TiO_2_, Fe oxides, Si, fullerenes, and nanotubes, either naked or coated, with more than 30 viruses, have been investigated [9,10,11,12,13,14,15]. As with the antibacterial action, the nanoparticles disrupt the activity of viruses by a range of mechanisms depending on the virus, the particle material, its size, and its coating. Figure 1 shows six of the antiviral mechanisms identified in the case of Ag nanoparticles [16,17].

Outside the cell, the nanoparticles can interact with the proteins that coat the virus and prevent it from docking with the cell (Figure 1a). Alternatively, the nanoparticles can interact with the cell membrane receptors that the viruses use for entry, thus preventing them from being internalized by the cell (Figure 1b). If the virus does enter the cell, the nanoparticles can hamper its transport through the cytoplasm so that it is unable to deliver the genome to the nucleus (Figure 1c). If the viral genome is released, the nanoparticles can interact with it and prevent it from being used to manufacture proteins (Figure 1d). If the virus can deliver a working genome, the nanoparticles can still prevent the assembly of new viruses by disturbing either the viral or cellular factors required to complete them (Figure 1e,f). Many of these mechanisms rely on Ag being in the form of nanoparticles.

However, the majority of the reported studies rely on wet methods to produce the nanoparticles, such as thermal decomposition [18,19], reduction of silver precursors [20,21], or using biological compounds [22,23]. The resultant hydrosols need to be stable, so surfactants are used, modifying the as-prepared surface of the nanoparticles [24,25]. Furthermore, in light of the recent pandemic, the coating of personal protective equipment (PPE) with Ag nanoparticles (or any material with antiviral activity) has become a trend in nanotechnology and bioscience. In the case of respiratory viruses, the majority of the studies are focused on coating facemasks. The coatings in the literature are diverse and go from soaking the substrates in the solution [26,27,28] or synthesizing the nanoparticles in the substrate [29,30] to producing nanofibers from the nanoparticles [31,32,33]. However, these are all two-step processes that result, as already stated, in nanoparticles with a modified surface. Gas-phase synthesis provides a method to coat textiles without using a stabilizing agent and enables the investigation of the interaction of the virions directly with the nanoparticles. To our knowledge, there are few studies using gas-phase synthesis [34,35,36,37,38], and they all rely on sputtering, a technique that produces continuous films, has low yield, and whose scale-up is not trivial.

The coating method used here, described in more detail in the following section, is a one-step process that passes a pure Ag nanoparticle aerosol through the cloth allowing the particles to be deposited on fibers throughout the volume. The method has been demonstrated previously to coat a range of textiles with Ag nanoparticles and assess their antibacterial performance against the bacteria *Staphylococcus aureus* and *Klebsiella pneumoniae* [39]. It was found that up to 99.96% of bacteria were neutralized with a silver loading, an order of magnitude less (that is, of the order of 10 mg/kg of textile) than in wet processes involving Ag hydrosols, which also require multiple processing stages.

It is, thus, of special interest to assess the antiviral performance of cloth treated in this way. In this report, we show that up to 98% of virions that have been exposed to nanoparticles deposited in the cloth are neutralized; that is, they are unable to infect cells after detaching from the treated cloth. The aerosol process used is cheap and easily scalable, and it is estimated that material costs would be less than EUR 0.05 for a typical face mask.

## 2. Materials and Methods

### 2.1. Ag Nanoparticle Synthesis and Cloth Coating

The Ag nanoparticles were produced by a spark source (Figure 2a) in which a pulsed high-voltage spark is generated between two electrodes made of the material required to form the nanoparticles [39,40]. The spark takes place in a flow of inert gas such as He or Ar and is normally produced by charging a capacitor until it reaches a sufficient voltage to initiate a spark and discharge via the plasma generated. Each spark produces a plume of supersaturated metal vapor from the electrodes that condenses into a nanoparticle aerosol in the inert gas. In principle, the particle size can be controlled by adjusting the spark power, the gas flow rate, and the tube aggregation length prior to deposition, but for the conditions used to prepare all our samples, the nanoparticle diameter was around 4 nm.

The device used in our experiments was a commercial source (VSP-G1) purchased from VS Particle B.V. (Delft, The Netherlands) (Figure 2b) and was operated typically at 2–10 W using Ar gas with a flow rate in the range 2–10 liters/minute (lpm). As shown in Section 3.1, the size variation of the particles within these conditions was small, and the particle diameter was typically 4 nm. Thus, although it was not possible to examine the effect of size on antiviral performance, it was found that the flow rate affected the crystallinity of the nanoparticles, as described in Section 3.1. Thus, for all depositions, the conditions used were a power of 5 W, which gave a well-stabilized spark and two flow rates, that is, 2 lpm and 8 lpm, to examine the effect of crystallinity on antiviral performance.

The aerosol was passed through different types of facemask cloths fixed perpendicularly to the flow, thus effectively working as a filter. For that purpose, the blank facemask was held by a commercial sample holder, as seen in Figure 2b. Figure 2c shows a typical sample after a deposition time of 30 min; it is estimated that the amount of Ag in this sample is 0.05 mg. This is a very small quantity, but the process is easily scalable and has been demonstrated to a level of 36 g/h (sufficient for over a ton of cloth per hour) from a single spark chamber while still maintaining a very high level of purity of the nanoparticles [42]. It was found in practice that depositions of at least an hour were required to produce a good antiviral performance, so all the cloths used for antiviral testing were coated for one hour or more.

### 2.2. Nanoparticle Characterisation

The gas phase nanoparticles were characterized by electron microscopy by collecting them on transmission electron microscopy (TEM) grids coated in lacy carbon placed parallel to the gas flow, thus picking up particles by diffusion. In this way, it was possible to collect sufficient nanoparticles to image and obtain size distributions in about 5 min. The images were obtained with the JEOL 1400 TEM (JEOL, Tokyo, Japan) at UCLM Albacete. After passing the aerosol through them, the cloth samples were imaged by a GeminiSEM 500 FESEM (ZEISS, Oberkochen, Germany) at the IRICA institute in Ciudad Real.

### 2.3. Determining the Antiviral Performance of Coated Cloth

An RNA enveloped virus (Usutu virus), and a non-enveloped virus (Murine norovirus) were used to assess the differences in antiviral performance of the coated facemasks to different proteins. The origin of the Usutu virus strain (GenBank accession number AY453411) used in this manuscript has been previously documented [43]. Murine norovirus was obtained by reverse genetics following the protocols described in [44] and using the infectious plasmid pT7:MNV-G 3′Rz, provided by Ian Goodfellow (University of Cambridge). In order to carry out the assays, the mask cloth samples were cut into 0.5 cm discs, onto each of which was added 3–4 µL of each viral sample. In order to favor the contact between virus particles and mask cloth surface, the samples were left to dry at 37 °C for 30–40 min. Infectious virus was then rescued by soaking the discs in 250 µL of cell culture media containing 1% (*v*/*v*) fetal bovine serum (FBS, Sigma Aldrich, Darmstadt, Germany), 100 units/mL penicillin–streptomycin (Gibco), and 1 mM Hepes in high glucose DMEM (Gibco). The supernatants of these resuspended cloths were then analyzed by 50% tissue culture infectious dose (TCID_50_) assays as previously described for both Usutu virus and murine norovirus [43,44]. The virus titres were determined by scoring the number of infected wells showing apparent cytopathic effect at day 5 post-infection and using the Reed and Muench method [45,46].

For the assays as a function of Ag nanoparticle deposition time (see Section 3.3), there were ten independent virus titer values, each coming from analyzing twice, each of 5 independent cloth discs. For the assays as a function of deposition conditions (see Section 3.4), the virus titre values were obtained from three independent cloth discs. The error bars shown in the plots were determined from the standard deviation of the results, and the confidence, as specified by the *p*-values obtained from one-way ANOVA tests, are discussed in Section 3.3 and Section 3.4.

## 3. Results

### 3.1. Characterisation of Gas Phase Nanoparticles

Figure 3 illustrates the characterisation of the gas-phase nanoparticles by TEM, with Figure 3a showing a typical image of a field of particles with diameters below 5 nm. The particle size distribution showed the usual asymmetric shape well described by a log-normal curve as plotted in Figure 3b for the same source conditions as the image in Figure 3a. There is little variation of particle size with source conditions, that is, power and gas flow rate, as shown in Figure 3c, and only at extreme conditions (for example, power = 13 W, gas flow rate = 0.5 lpm) is there a significant change in size. For all samples produced for antiviral assays, it can be assumed that the particle size was about 4 nm.

Figure 3d,e show high-resolution TEM images of individual nanoparticles, revealing lattice planes, along with the corresponding FFT patterns displayed in the yellow boxes. The particle in Figure 3d is a single crystal showing just one orientation of 110 planes with an interlayer spacing of 0.28 nm corresponding to metallic Ag. Figure 3e reveals a circle of spots corresponding to the lattice spacing (0.23 nm) of the (111) planes of metallic Ag, indicating a polycrystalline particle with around 20 nanocrystallites. There were also individual nanoparticles found with bi- and tri-crystals; thus, the structure varies from single crystals through to polycrystalline, but in all cases, with no evidence of silver oxide within the particle. In addition, there is no evidence of Ag oxide at the surface of the nanoparticles, which would show up as a lower density shell in the images. It was found that at low gas flow rates, there was a significantly higher proportion of nanoparticles with a single crystal structure than at high flow rates, at which most nanoparticles were polycrystalline presenting (111) facets at the surface. We believe this is due to the faster cooling rate at the higher gas flow, and as we show in Section 3.4, the crystallinity of the nanoparticles has an effect on the antiviral performance.

### 3.2. Characterisation of Nanoparticles Deposited in Cloth

The aerosol was passed through the cloth used to make surgical facemasks for periods ranging from 30 min to 4 h and imaged by SEM. Figure 4a shows a low magnification image of the cross-section of the mask, with a total width of about 450 µm, consisting of three layers, that is, a front cloth sheet, a central filter of width around 180 µm, with a higher density weave and a rear cloth sheet. The silver signal from the elemental analysis using energy-dispersive X-ray analysis (EDX) from a coated mask for the three regions is shown in Figure 4b and reveals that not many nanoparticles are trapped in the front cloth sheet, with the vast majority residing in the filter and no detectable signal in the rear cloth sheet. In this diagram, the magnitude of the Ag signal is plotted as a percentage of the total elemental signal in all elements (mainly carbon). The variation of the Ag EDX signal within just the filter section is plotted as a function of depth in Figure 4c, and it is seen to be a reasonable fit to an exponential curve shown by the red dashed line. It is evident from Figure 4b,c that the nanoparticles are almost entirely stopped within the filter.

Figure 5a,c show SEM images at different magnifications of the deposit on a single fiber in the filter after a deposition time of 30 min with a power of 5 W and a flow rate of 8 lpm. Figure 5b,d show similar images after a deposition time of 4 h with the source operated under the same conditions, and in all cases, the fibers are well covered with nanoparticles. Coating for 30 min produces a mixture of individual particles and agglomerates, while after 4 h, a continuous, highly porous nanoparticle deposit is observed (Figure 5b,d).

To demonstrate that the agglomeration is occurring on the fibers and not in the beam (as has been sometimes reported for this type of synthesis [47,48]), Figure 6 shows an SEM image of a lacy carbon TEM grid after passing the aerosol over it for 5 min in a similar manner to the samples prepared for the TEM images. Here it is observed that most of the coating is made of individual nanoparticles. Furthermore, shown in the figure, approximately to scale [49], is a schematic of an Usutu virion used in this study; it is clear that even at low deposition times, a virus particle landing on fiber will come into contact with several Ag nanoparticles.

### 3.3. Antiviral Performance of Coated Cloth vs. Nanoparticle Deposition Time

Initially, as a test of the effect of deposition time, that is, the density of nanoparticles within the cloth, we determined the antiviral performance of cloths coated for one hour on one side, two hours on one side, and an hour on each side for both the murine norovirus (MNV) and Usutu virus (USUV). For all depositions, the nanoparticle source was operated at a power of 5 W and a flow rate of 8 lpm, that is, with the same conditions as used for the data in Figure 3. The results are plotted in Figure 7, and it is seen that for both viruses, the best performance is obtained for a two-hour deposition on one side, giving a 20× and a 60× reduction in infectious virus in the case of MNV and USUV, respectively. The *p*-values obtained from one-way ANOVA tests indicate a significant decrease, compared to the control, in infectious virus for all treatments in the case of MNV (*p* < 0.0001 for 1 + 1 and 2 h and *p* < 0.05 for one hour). For the USUV assays, only the two-hour deposition gave a significant decrease (*p* < 0.001); however, the headline result remains, that is, the best performance is obtained by a two-hour deposition on one side for both viruses.

As can be ascertained with reference to Figure 5, in the case of the one-hour deposition, most of the nanoparticles will be aggregated on the cloth fiber, while after two hours, all the particles will be in aggregates. Thus, the best performance is obtained in the case of agglomerated films, which is counterintuitive and is discussed in Section 4. Increasing the deposition time to four hours does not improve the antiviral performance, which is not surprising since the morphology of the surface to which the virus particles are exposed is similar to the one for a two-hour deposition.

### 3.4. Antiviral Performance of Coated Cloth vs. Source Conditions

It was demonstrated in Section 3.1 that the source conditions have little effect on the nanoparticle size distribution, but the gas flow rate does affect the crystallinity; that is, low flow rates tend to produce single crystal nanoparticles while high flow rates tend to produce polycrystalline ones. We examined whether this has an effect on antiviral performance by applying the assay to mask cloth containing nanoparticles that had been deposited with a flow rate of 2 lpm and 8 lpm with a power of 5 W and a deposition time of one hour in both cases. Note that the amount of Ag deposited depends on the power only and is the same for both samples. The results for murine norovirus and Usutu virus are shown in Figure 8, and as before, the results are the average of several assays with the error bar indicating the spread among the results. In the case of the murine norovirus (Figure 8a) there is a clear trend of increased performance with increased flow rate, indicating that polycrystalline nanoparticles have a stronger interaction with the viral proteins than single crystals. The associated *p*-values (*p* < 0.0001 for 8 lpm and *p* < 0.05 for 2 lpm) indicate that for both treatments, the level of infectious virus is significantly lower than in the control. For the Usutu virus (Figure 8b), although the average values show the same trend as for the murine norovirus, the error bars and associated *p* values prevent us from making the same claim with confidence.

## 4. Discussion

Within the error bars, we found that the antiviral performance for the two viruses tested was similar. In the case of the murine norovirus, the outer layer is composed of VP1 protein dimers self-assembled into a spherical particle to form the capsid, whereas, for the Usutu virus, the Ag nanoparticles would interact with the USUV-E membrane proteins. In addition to these differences, the two viruses are from different families; that is, Flaviviridae for USUV and Caliciviridae for MNV. This indicates that the action of the Ag nanoparticles is generic and does not depend on the detailed structure of the outermost proteins or, indeed, the family of the virus. Thus, we expect the antiviral action of the nanoparticles to be manifested for viruses in general, in particular, for the SARS-CoV-2 virus, which this research is aimed at.

The antiviral performance of the films was found to increase with deposition time, reaching a maximum when the cloth fibers were totally covered by films of agglomerated Ag nanoparticles, and increasing the deposition time further produced no detectable increase in the effectiveness. Thus, the important criterion is that the fibers are completely covered and not that the coatings are composed of isolated nanoparticles, in contrast to what we originally expected. Thus, the virus particles that come into contact with the Ag deposits are able to pull nanoparticles off them and carry these into the cell culture. This implies that the agglomerates are loosely bound in an aqueous environment, and the interaction with the proteins is sufficiently strong to separate nanoparticles from the agglomerate.

Although we were unable to investigate the effect of particle size on the antiviral performance, we were able to change the crystallinity of the nanoparticles from predominantly single crystals to predominantly polycrystals. The nanoparticles are sufficiently large to be expected to have the bulk fcc crystal structure, and the surface plane with the minimum specific surface energy of an fcc crystal is the (111) face. Previous studies synthesizing pure Ag nanoparticles with a spark ablation source also report an fcc structure [50]. It has been shown, however, that the minimum surface energy of an fcc nanoparticle, which is produced by a compromise between minimizing the total surface area and the specific energy of faces, is a cuboctahedron, which is a mixture of (111) and (100) faces [51]. On the other hand, metastable polycrystalline nanoparticles present almost exclusively (111) faces but, in addition, will have a higher density of surface defects. The more densely packed (111) faces would be expected to have a stronger interaction with the viral proteins, as would the defect sites. This suggests that the virus particles have an enhanced ability to pull polycrystalline nanoparticles off the films, which explains the enhanced antiviral performance of the polycrystals, at least in the case of the murine norovirus.

In terms of practical applications, this research is aimed at identifying any benefits of coating face masks, widely used in the current pandemic, with Ag nanoparticles. We should first note that the filters in untreated masks are highly effective at stopping nanoscale particles, as demonstrated in Figure 4. However, in the coated masks, up to 98% of virus particles that do pass the filter are neutralized; that is, the mask becomes up to fifty times more opaque to the infectious virus, and given that it only takes a single virus particle to start an infection, this is an important factor. In addition, there is, as far as we are aware, no information available on what happens to the viral load in masks after use. A recent study has shown that, typically, a face mask contains about 2500 virus particles after 2–3 h use by an infected patient [52]. There is clearly a potential for virus particles to survive and be released back into the environment, but with the treatment described, up to 98% of these particles will have been rendered non-infectious. Before applying this technology, however, it must also be demonstrated that the nanoparticle films are stable against the airflow produced by breathing and are not ingested directly into the lungs.

## 5. Conclusions

In conclusion, we have studied the effectiveness of 4 nm diameter Ag nanoparticles deposited onto facemask cloth in neutralizing murine norovirus and Usutu virus particles. For optimum coating conditions, the coatings render 95–98% of virus particles exposed to the films non-infectious. It was found that the best performance was obtained with films of agglomerated nanoparticles, and, in the case of murine norovirus, polycrystalline nanoparticles perform better than single crystals.

## Figures and Tables

**Figure 1 nanomaterials-12-02662-f001:**
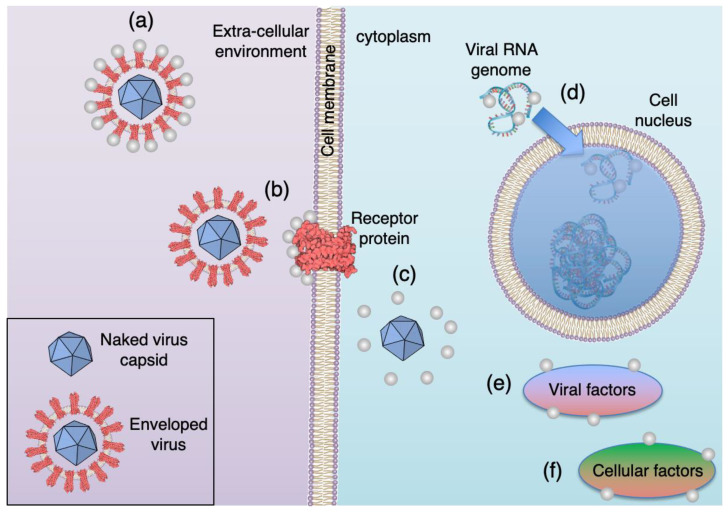
Antiviral mechanisms of Ag nanoparticles. (**a**) Ag nanoparticles interact with the viral envelope and/or viral surface proteins and prevent virus from docking with the cell. (**b**) Ag na-noparticles interact with cell membrane receptors and block viral penetration. (**c**) Ag nanoparti-cles block cellular pathways of viral entry preventing the genome reaching the nucleus (**d**) Ag nanoparticles interact with the viral genome and prevent it from being used to manufacture pro-teins. (**e**,**f**) Ag nanoparticles interact with viral and cellular factors necessary for viral repli-cation. Reproduced with permission from [17]. Copyright 2022, Wiley.

**Figure 2 nanomaterials-12-02662-f002:**
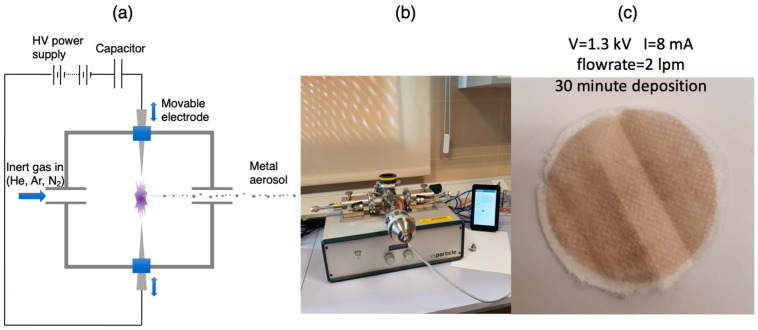
Nanoparticle synthesis and textile coating. (**a**) Production of nanoparticle aerosol using a spark source. (**b**) the device used, which was the VSP-G1 source produced by VSParticle B.V. The cloth to be coated is trapped in a commercially available chamber and acts as a filter for the nanoparticles. (**c**) A typical coating on a piece of facemask material after 30 min with the source parameters shown. (**a**) Reproduced with permission from [41]. Copyright 2021, John Wiley and Sons.

**Figure 3 nanomaterials-12-02662-f003:**
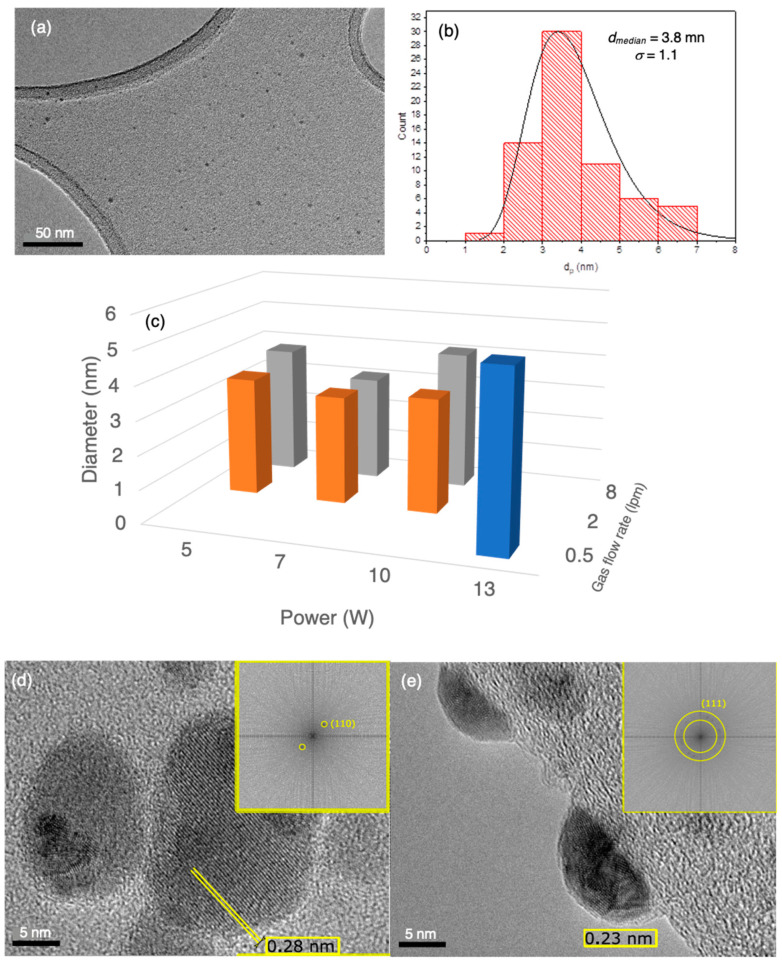
Characterisation of gas phase nanoparticles. (**a**) Typical field of Ag nanoparticles produced with a power setting of 5 W and a gas flow rate of 8 lpm indicating a particle diameter below 5 nm. (**b**) Size histogram for the same source conditions fitted to a log-normal distribution, with the median diameter and standard deviation indicated. (**c**) Variation of the median diameter with the power and gas flow rate in the source. (**d**) TEM image of an individual nanoparticle at a lacy carbon edge with a monocrystalline structure as indicated by single pair of spots corresponding to the Ag (110) planes in the FFT shown in the inset. (**e**) TEM image of an individual nanoparticle at a lacy carbon edge showing a polycrystalline structure as indicated by a ring of spots within the yellow circles corresponding to the Ag (111) planes in the FFT shown in the inset.

**Figure 4 nanomaterials-12-02662-f004:**
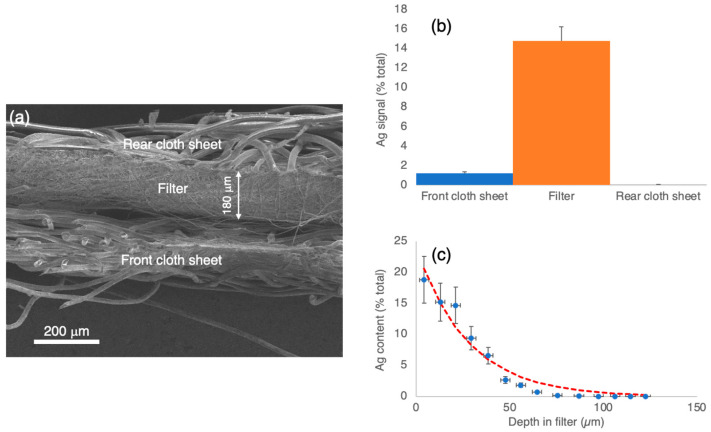
Distribution of Nanoparticles within the masks. (**a**) Low magnification SEM image of the mask cross section showing the three-layer structure with a front cloth sheet, the high-density weave filter and the rear cloth sheet. (**b**) EDX signal from Ag in a coated mask, expressed as a % signal from all elements, in the front sheet, filter and rear sheet. (**c**) EDX Ag signal as a function of depth in the filter of a coated mask fitted to an exponential curve (dashed line). It is evident that virtually no nanoparticles are transmitted through the mask.

**Figure 5 nanomaterials-12-02662-f005:**
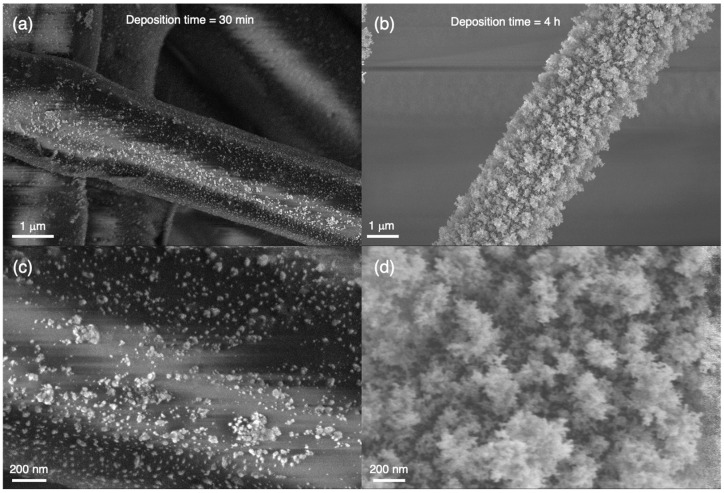
SEM images of Ag nanoparticle deposits in the filter section. (**a**) Image of nanoparticles on a single fibre in the filter after a deposition time of 30 min with a power of 5 W and a flow rate of 8 lpm. (**b**) Image of nanoparticles on a single fibre on the filter after a deposition time of 4 h. (**c**,**d**) Similar images at higher magnification. The coating after 30 min consists of a mixture of individual nanoparticles and agglomerates while after 4 h, all particles are in agglomerates.

**Figure 6 nanomaterials-12-02662-f006:**
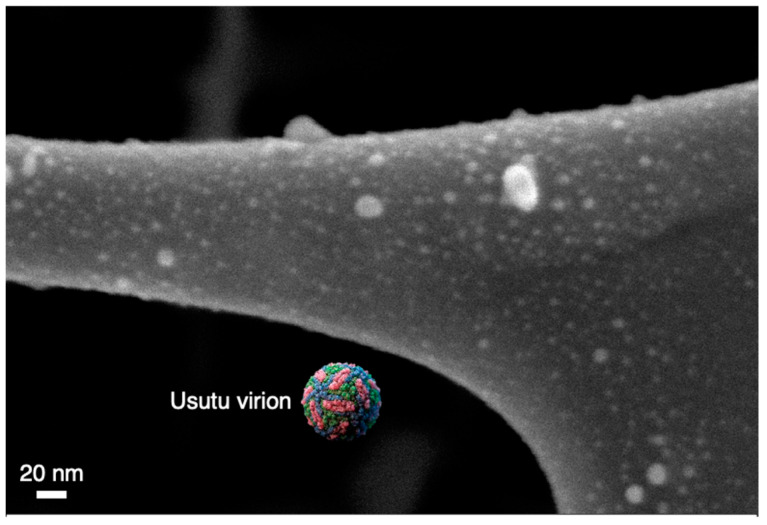
SEM image of Ag nanoparticles on a lacy carbon TEM grid. High-magnification SEM image of a 5-min deposit on a lacy carbon TEM grid showing that the majority of the coating is individual nanoparticles.

**Figure 7 nanomaterials-12-02662-f007:**
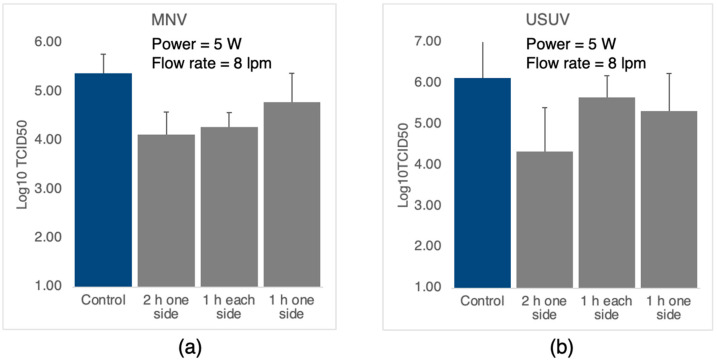
TCID50 assays for different coating times. (**a**) Cloths exposed to murine norovirus. (**b**) Cloths exposed to Usutu virus. The nanoparticle source conditions were power = 5 W, flow rate = 8 litres/min. The number of infectious virus particles has been reduced by a factor of 20× for MNV and 60× for USUV.

**Figure 8 nanomaterials-12-02662-f008:**
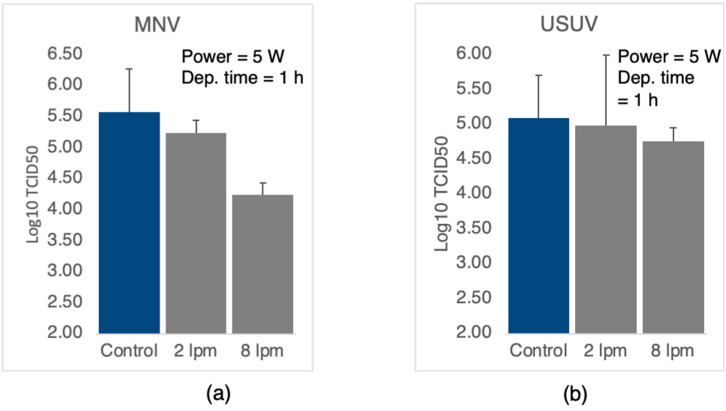
TCID50 assays for different flow rates. (**a**) cloths exposed to murine norovirus. (**b**) Cloths exposed to Usutu virus. The nanoparticle source conditions were power = 5 W, and the flow rate was 2 or 8 litres/minute with a deposition time of one hour.

## Data Availability

The data is available on request from the corresponding author.

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
