# Peer review of "Effectiveness of Silver Nanoparticles Deposited in Facemask Material for Neutralising Viruses"

_nanomaterials, 2022, doi:10.3390/nano12152662_

Round 1

Reviewer 1 Report

Raúl López et al. (submission nanomaterials-1808280) investigate the effectiveness of cloth used for facemask coated with silver nanoparticles through a dry aerosol process. The work demonstrate that silver nanoparticles embedded in masks can neutralize murine norovirus and usutu virus particles.

It is an interesting article in general, particularly in view of practical applications in the current COVID19 pandemic situation. Yet, the introduction does not give a good overview of the scientific background, especially about the possible coating methods. The choice of these two viruses is not justified and a rationale should be provided upfront.

The experimental design could be elucidated in detail from the beginning (es. the select different flow rates and times). The authors could have been bolder in claims and try to better value their results. In this reviewer’s view, in fact, the feeling is that the work could have been expanded more, especially for a much deeper discussion and highlighting the impacts. At the very least it is requested to update the references with some newer ones and to expand the reference list to 50 or more since this filed (Ag nanocoatings) is very established  . Apart from this, it is a well written, timely, and useful paper per se, which should be published after some  revisions are addressed.

Specific comments:

Apart from the points made earlier, some other specific aspects are pointed out:

Page 3, line 86: explain better the potential advantages of dry aerosol process over others in terms of manufacturing and/or from application viewpoint. This work deserves to be better framed in the reference research field  

Page 6, line 242: In figure 4b, please remove minor grid lines also to resolve overlap of “b” mark with the line in the chart. Same advise goes for all figures , e.g. Figure 7 or 8. In Figure 8 axes are missing.

Page 7, line 260: specify the values for W and Flow Rate for the deposition of 4 hours, or whether they were the same as before.

Page 8, line 322: In figure 7b there is one too many “I” into the word “hour”.

Is the obtained reduction  statistically significance?

Page 9, line 352: Even though the error bars were not promising, in Figure 8 authors are encouraged to report also the graph for the USUTU virus. Here too, there is not marking about statistical analysis / significance (by P-value, t-test or other methods). Labeling and text does not seem adequate and a proper font/size should be adopeted throughout while ensuring readability and proportions are safeguarded relative to each graphics

About the results:

-          Why has the test of the effect of deposition time not been determined also for the condition “2 hours each side”?

-          There is not statistical analysis of results in the Results and Figures. In materials and methods there is in fact no sign of statistical approach. It needs to be added.

Reviewer 2 Report

Authors have done intensive works on Effectiveness of Silver Nanoparticles Deposited in Facemask Material for Neutralising Viruses. It is well present. 

 Please clarify what is 'section 3.1'  in line 137 at section 2.1 and 'section 3' in line 226. 

The order of presentation should be justified. The result should be later than material method. It is difficult to follow. 

Round 2

Reviewer 2 Report

Authors have improved the manuscript. 

1. Please use alignment to justify the all the text in paragraph of word file.

2 Citation in the left column in the first page

Martín, R.L.; Rodrigo, I.;  Ballesta, C.; Arias, A.; Mas, A.; Burgos, B.S.;    Normile, P.S.;  De Toro, J.A. and  Binns, C. Effectiveness of Silver Nanoparticles Deposited in Facemask Material for Neutralising Viruses. Nanomaterials

3 line 456-471

Formal Analysis: R.L.M, A.A, A.M, B.S.B and C.B.;  Investigation: R.L.M., I.R, C.B., A.A, A.M. and B.S.B; writing—original draft preparation: C.B.; writing—review and editing:  R.L.M, I.R, A.A, A.M, B.S.B, J.A.D.T, P.S.N. and C.B. Obtaining Funding: A.A, A.M, J.A.D.T, P.S.N. and C.B.  All authors have read and agreed to the published version of the manuscript.

4 THe new reference 16 should be 

Rai, M.; Deshmukh, S.D.; Ingle, A.P.; Gupta, I.R.; Galdiero, M.; Galdiero, S. Metal nanoparticles: The protective 520 nanoshield against virus infection. Crit. Rev. Microbiol.  2014, 42, 46–56.